# Unsupervised Analysis of Small Molecule Mixtures by Wavelet-Based Super-Resolved NMR

**DOI:** 10.3390/molecules28020792

**Published:** 2023-01-13

**Authors:** Aritro Sinha Roy, Madhur Srivastava

**Affiliations:** 1Department of Chemistry and Chemical Biology, Cornell University, Ithaca, NY 14850, USA; 2National Biomedical Center for Advanced ESR Technology, Cornell University, Ithaca, NY 14850, USA

**Keywords:** NMR, shift spectra, wavelet packet transform, automated small molecule mixture analysis

## Abstract

Resolving small molecule mixtures by nuclear magnetic resonance (NMR) spectroscopy has been of great interest for a long time for its precision, reproducibility, and efficiency. However, spectral analyses for such mixtures are often highly challenging due to overlapping resonance lines and limited chemical shift windows. The existing experimental and theoretical methods to produce shift NMR spectra in dealing with the problem have limited applicability owing to sensitivity issues, inconsistency, and/or the requirement of prior knowledge. Recently, we resolved the problem by decoupling multiplet structures in NMR spectra by the wavelet packet transform (WPT) technique. In this work, we developed a scheme for deploying the method in generating highly resolved WPT NMR spectra and predicting the composition of the corresponding molecular mixtures from their ${}^{1}$H NMR spectra in an automated fashion. The four-step spectral analysis scheme consists of calculating the WPT spectrum, peak matching with a WPT shift NMR library, followed by two optimization steps in producing the predicted molecular composition of a mixture. The robustness of the method was tested on an augmented dataset of 1000 molecular mixtures, each containing 3 to 7 molecules. The method successfully predicted the constituent molecules with a median true positive rate of 1.0 against the varying compositions, while a median false positive rate of 0.04 was obtained. The approach can be scaled easily for much larger datasets.

## 1. Introduction

Identification of the components of small molecule mixtures is a crucial and challenging step in the research and development activities in the pharmaceutical drug discovery [1,2,3], metabolomics [4,5,6], natural product synthesis [7,8,9], food quality control [10,11], and environmental sciences [12,13]. Different types of nuclear magnetic resonance (NMR) spectroscopic methods, high-performance liquid chromatography (HPLC), and mass spectrometry (MS) are widely used across the associated industries for this purpose. The main advantages of NMR over the other techniques are that (1) its results are highly reproducible, (2) it requires very little sample preparation effort, and (3) it is a nondestructive method [14,15,16]. However, its relatively poor resolution and sensitivity often make NMR an essential, but non-exhaustive analytic tool [5,9]. While recent developments for sensitivity improvement have largely been successful [17,18,19], limited progress has been made towards achieving the desired resolution. This is primarily due to the limited range of chemical shift windows (∼10 ppm) and overlapping resonance lines of small molecules. It is possible to enhance the resolution in homonuclear decoupled ${}^{1}$H NMR spectroscopy by producing pure shift spectra [20,21,22,23,24]. The technique has failed to gain wide applicability owing to its experimental complexity and poor sensitivity [20,25]. While multi-dimensional NMR can improve the resolution by revealing some of the overlapped components, it comes at the cost of a high signal acquisition time and experimental noise, which makes it unsuitable for automated high-throughput studies. To overcome the limited chemical shift window of 1D ${}^{1}$H NMR, pseudo-2D NMR methods are employed using diffusion coefficients, relaxation parameters, and other suitable molecular parameters for spectral simplification and differentiation in NMR [26,27,28,29]. However, the accuracy of the extraction of such molecular parameters and, hence, the efficiency of separating spectral components for a mixture depends heavily on the molecular size distribution, the extent of spectral overlapping, and the magnetic properties of the molecules in a mixture [29]. Therefore, these methods are often complementary to each other and cannot be applied without user interference or prior knowledge about the mixture or data pre-processing, which adds further limitations to the applicability of the methods [26,30]. In order to resolve the problem theoretically, the maximum entropy method has been used in converting NMR to pure shift spectra by deconvolution [31,32]. One of its major drawbacks is the requirement of prior knowledge about the scalar coupling patterns and the coupling constants, which is reasonable for ${}^{13}$C NMR, but unsuitable for ${}^{1}$H NMR spectroscopy. Apart from this, a series of spectral analysis tools has been developed, which include peak matching strategies [33,34,35], spectral editing [36,37], similarity measure [38,39], and deep-learning-based tools [40,41,42], for identifying small molecule mixture constituents from the corresponding NMR spectra. However, those applications can be seldom generalized, often suffer from low reliability, and/or require extremely large and specifically designed training datasets. As a result, none of the methods are suitable for high-throughput analysis of small molecule mixtures using ${}^{1}$H NMR as the primary tool.

In a recent work, we showed that the wavelet packet transform (WPT) can work as a multi-resolution signal processing tool in transforming an ${}^{1}$H NMR to a pure shift spectrum [43]. Successive decomposition of a spectrum by WPT yields pairs of approximation and detail components at each level, which contain some of the low- and high-frequency spectral features from the chemical shift domain, respectively. The approximation component produced at the final level of decomposition of an ${}^{1}$H NMR spectrum produces only singlet structures, while the multiplet structures are transferred to the various detail components. We illustrate that the former can be used to calculate a simple pure shift spectrum, and the robustness of the WPT-based NMR spectral analysis method against a significant level of noise has been established [43]. An overview of the wavelet transform theory is provided in the Appendix A.

Automating the task of molecular identification in the study of metabolites and other small molecule mixtures without a priori knowledge remains a major challenge, especially using 1D NMR as the primary analytical tool [44,45,46]. In absence of an automated mixture analysis, the study time increases significantly, while the accuracy of the analysis varies widely based on the user inputs and interpretations, as well as the nature of the molecular mixtures [44,47,48]. In this work, we developed an automated method for predicting molecular compositions from the corresponding 1D ${}^{1}$H NMR spectra without a priori knowledge and demonstrate its applicability across a wide range of molecules. The problem of the automated identification of mixture components from the corresponding 1D ${}^{1}$H NMR spectra can be divided into two parts: (1) predicting the number of molecules in a mixture and (2) predicting their identities. For this purpose, we created an extensive database of 1000 augmented NMR spectra of molecular mixtures, each containing 3 to 7 spectra of the constituent molecules. A library of WPT shift NMR was created from the 500 MHz NMR spectra of 74 molecules. The mixed NMR spectra were analyzed in an automated fashion by implementing a four-step algorithm. The algorithm in its first two steps calculates a WPT shift spectrum from an NMR spectrum and obtains a potential molecular composition by matching the peaks in the shift spectrum with those of the spectral library. Next, the composition is optimized by employing a gradient descent method to minimize the mean-squared error in predicting the WPT shift spectrum of the mixture. The top 15 entries from the potential composition are forwarded to the next step, where another gradient-descent-based minimization in predicting the WPT spectrum of the mixture produces the final list of molecules.

In analyzing the performance of the method, we used the true positive and false positive rates, which represent the number of accurate and false predictions with respect to the actual compositions of a molecular mixture, respectively. After the first optimization step, we obtained an average true positive rate of 1.0, while the average false positive rate was very high (0.3). This elimination step removed the molecules (choices) with zero or very low probability to be present in the composition from the probable list. Among the remaining choices, the top entries by their calculated probability of existence describe the true compositions for all the cases in the augmented dataset. In fact, we observed that, for mixtures with 3 to 4 molecules, a true positive rate of 1.0 could have been obtained considering only the top 6 to 8 entries, respectively. Therefore, selecting the optimal number of entries from the potential list of molecules without a priori information requires a second optimization. In this identification step, the top 15 entries obtained at the end of the elimination step were optimized by another gradient descent method, which resulted in a median true positive rate of 1.0, while reducing the false positive rate to 0.04 for the analysis.

## 2. Results and Discussion

As a benchmark, we analyzed the dataset of mixed spectra by matching them with the pure NMR spectra of the individual molecules, which is the most commonly used strategy at present [33,34]. From the summary of the analysis shown in Figure 1, it can be seen that the average true positive rate is ∼0.7 for the entire dataset, as well as for the mixtures with different numbers of constituent molecules in them. Both subplots in the figure show large variations in the true positive rate, which demonstrates the high uncertainty involved in the analysis. The false positive rate for all the cases was equal to 0. It should also be noted that this kind of direct matching may not be feasible for much larger libraries than the one with 74 molecules used in this work.

The results obtained in Step (3) of our scheme (Figure 2) are summarized in Figure 3. At this stage of our analysis, a median true positive rate of 1.0 was obtained for the entire spectral dataset. This observation demonstrates the robustness of the WPT shift representation of a NMR spectrum and its ability to enhance the resolution [43]. However, the impressive true positive rate was associated with a very high false positive rate across all the cases, with a median value of 0.3. Both the average false positive rate and its variations increased with the size of the mixtures or the increasing complexity of the corresponding spectra.

Looking at the individual analyses and the corresponding mixture compositions, we noticed that, while ∼30 molecules were present in each prediction on average, leaving a few outliers, the top 6 to 15 entries by their probability of existence contained all the components of the mixtures. Hence, in the final step of our spectral analysis scheme, we employed another optimization, which used the top 15 entries from a predicted molecular composition in Step (3). This step resulted in a massive reduction in the false positive rate from 0.3 in Step (3) to 0.04, shown in Figure 4, while the true positive rate remained mostly unaffected, except for the case with molecular mixtures comprising seven molecules. Even for the latter case, we obtained a median true positive rate of 1.0 with a standard deviation of 0.08.

For visualization purposes, we plot the predicted NMR spectra from the component analysis for a set of four representative cases and compared those with the corresponding mixed NMR spectra, shown in Figure 5. The descriptions for the representative mixtures are given in Table 1. In Figure 5A, a simple visual inspection could remove the false entries: astaxanthin, indolelactive acid, and L fucose. The probable cause for their inclusion in the final prediction was partial overlap between the molecular and mixed NMR spectra. In contrast, the top four molecules of the prediction in Figure 5B corresponded to the composition of the molecular mixture 23. Two of the three false positives, nicotinuric acid and linalyl acetate, could be discarded by visual inspection, and the probability of the third one, sulcatone, is less than half of that of 1,8-cineole. Figure 5C illustrates a similar analysis for a mixture containing five molecules, predicted by the top five molecules in the analysis. An easy elimination of the false positives, nicotine and catechin, by visual inspection is achievable in this case as well. In the last example, Figure 5D, the top seven molecules in the prediction capture all six molecules in the corresponding mixture. The false positive, shikimic acid, shows up in this list because of its high degree of overlap with the mixed spectrum. However, the missing peak in the mixed spectrum between 7 and 8 ppm could be used to remove it from the predicted composition. The rest of the false positives, nicotine and sucrose, can be eliminated by visual comparison of the actual and the predicted spectra. Our method’s performance summary is given in Table 2.

## 3. Materials and Methods

### 3.1. NMR to WPT Spectral Conversion

Recently, we utilized the properties of wavelet transforms for two different types of magnetic resonance spectroscopies, extracting hidden features from continuous wave electron spin resonance (cw-ESR) spectra [49] and producing highly resolved shift spectra from standard ${}^{1}$H NMR spectra [43]. In the latter case, the input NMR spectrum is decomposed by WPT, yielding a pair of approximation and detail components, effectively separating the low- and high-frequency components in the chemical shift domain. The term frequency is used in a generic sense here, and for a particular multiplet structure, the decomposition is continued until the derived approximation component produces a broad singlet encompassing the spectral domain. Subsequently, the multiplet in the original NMR spectrum is replaced by the peak position and height of the approximation component in obtaining the shift spectrum. This process is continued for an entire spectrum to obtain the corresponding WPT shift spectrum, while the approximation component itself is called the WPT spectrum. An example of such a spectral conversion for glutathione is shown in Figure 6.

A detail description of the method and wavelet transforms can be found in [43,50].

### 3.2. Spectral Library and Augmented Dataset Creation

We built a spectral library with the NMR spectra of 74 small molecules. Both experimental and predicted NMR spectra were used in the library based on data availability from a peer-reviewed publication [40] and the Human Metabolome Database [51]. The corresponding WPT spectral library for the molecules was computed using the Daubechies-9 (Db9) wavelet, and a full reduction of all multiplets to singlets in a spectrum was used as the criterion to select the optimal decomposition level. We mixed the NMR spectra of 20 molecules from the library to create an augmented dataset of 1000 spectra, shown in Figure 7. Only 20 molecules were chosen in creating the augmented spectra for two reasons, (1) setting a known list of true negatives and (2) making the analysis realistic, because the mixtures in most applications usually contain structurally related molecules. Each mixed spectrum was calculated by mixing 3 to 7 randomly selected molecules’ NMR spectra in varying proportions from 0.15 to 0.4. In creating a mixed spectrum, the component spectra were added in such a way that the number of data points in the mixed spectrum equaled the mean length of the individual spectra [43].

### 3.3. Automated Spectral Analysis Algorithm

The algorithm used can be seen as a four-step process, (1) the conversion of an NMR spectrum to its WPT and WPT shift versions, (2) matching peaks with the WPT shift NMR library and producing a sorted list (L I) of potential components, (3) optimizing L I to L II by applying a linear gradient descent algorithm, and (4) optimizing the top 15 entries of L II to produce the final prediction of the molecular composition of a mixture. The scheme is summarized in Figure 2. Both optimization steps used linear gradient descent algorithms, but the targets (*Y*) were taken to be WPT shift spectra for Step (3) and the WPT spectra for Step (4). WPT shift spectra are much simplified versions of the corresponding WPT spectra, where only the peak positions and peak heights from the latter are used [43]. The design matrices (*X*), whose columns correspond to the potential molecules in L I or L II, were constructed from the intersections of the chemical shift values from *Y* and the WPT shift/WPT spectral intensities for the individual molecules. The cost function, *J* [52], and the gradient descent minimization are given by
(1)$$\begin{array}{cc} J(\Theta ) & =\sum {\left(Y-X\cdot \Theta \right)}^{2}/m\\ \Theta_{i+1} & =\Theta_{i}-\alpha \;\nabla_{\Theta }\;J\left(\Theta_{i}\right) \end{array}$$
where *m* is the dimension of *Y*, Θ contains the probabilities for a set of molecules to be present in a mixture, $\nabla_{\Theta }\;J\left(\Theta \right)$ represents the gradient of the cost function, and α is the learning rate. For our method, the target *Y* and the design matrix *X* are described in Table 3. The chemical shift and intensity values from the WPT shift spectrum (in Step 3) and the WPT spectrum (in Step 4) of an experimental NMR spectrum of a small molecule mixture were used to define $Y_{1}$ and $Y_{2}$. Each of the columns in *X* corresponds to the molecules in our database, $\left({\mathrm{Molecule}}_{1},\ldots ,{\mathrm{Molecule}}_{n}\right)$. The matrix elements, $x_{ij}$, were calculated by matching the WPT shift (Step 3) and WPT (Step 4) spectrum intensity of ${\mathrm{Molecule}}_{j}$ to the chemical shift value of $\delta_{i}$, assigning $x_{ij}=0$ if $\delta_{i}$ fell outside of the spectral domain of ${\mathrm{Molecule}}_{j}$.

The value of α in Equation (Equation 1) was chosen to be 0.1 in Step (3), and for each iteration in Step (4), α was selected randomly from a uniform distribution in the range of 0.01 and 0.1. The steps in the algorithm are summarized as follows:Calculate WPT and WPT shift spectrum from an NMR spectrum;Match the WPT shift spectrum with the WPT shift spectral library:(a)*p* = count the number of matches for each molecule in the library;(b)The probability for a molecule to be in the mixture = p/the number of peaks in the WPT shift spectrum of the molecule;(c)Continue for all the molecules in the library, and short-list the ones with non-zero probabilities into the list, L I.Optimize the short-listed molecules by a gradient descent method:(a)Define the WPT shift NMR spectrum of a molecular mixture as the target variable, $Y_{1}$;(b)Create a design matrix, $X_{1}$, from the intersection of the chemical shift values from $Y_{1}$ and the intensities of the spectra for the molecules in L I;(c)Minimize $\sum {\left(Y_{1}-X_{1}\cdot \Theta \right)}^{2}/n_{1}$, where $n_{1}$ is the dimension of $Y_{1}$ and Θ is the probabilities associated with the molecules in L I, using a gradient descent method with a learning rate, α = 0.1;(d)An optimized list of molecules, L II, associated with non-zero probabilities is obtained.The top 15 entries from L II are used as the input to another optimization step:(a)Define the WPT NMR spectrum of a molecular mixture as the target variable, $Y_{2}$;(b)Create a design matrix, $X_{2}$, from the intersection of the chemical shift values from $Y_{2}$ and the intensities of the spectra for the molecules in L II;(c)Minimize $\sum {\left(Y_{2}-X_{2}\cdot \Theta \right)}^{2}/n_{2}$ using a gradient descent method with the learning rate chosen randomly from a uniform distribution between 0.01 and 0.1;(d)An optimized list of molecules associated with probabilities greater than 0.1 is obtained.

We used thew true positive and false positive rates as the metrics in evaluating the performance of our spectral analysis method, defined as follows:$$\begin{array}{cc} \; & \mathrm{True}\;\mathrm{positive}\;\mathrm{rate}=\frac{\mathrm{True}\;\mathrm{assignments}}{\mathrm{Actual}\;\mathrm{composition}}\\ \; & \mathrm{False}\;\mathrm{positive}\;\mathrm{rate}=\frac{\mathrm{False}\;\mathrm{assignments}}{\mathrm{Spectral}\;\mathrm{library}\;-\;\mathrm{True}\;\mathrm{assignments}} \end{array}$$

### 3.4. An Example of How the Scheme Works

For an illustration, we selected the molecular mixture 82, which contains 7 molecules: lactic acid, caffeine, citral, geraniol, 2-heptanone, furfuryl alcohol, and benzyl acetate. The NMR spectrum of the mixture is shown in Figure 8, followed by the analysis. The analysis started with the calculation of the WPT spectrum ($Y_{2}$) such that all multiplets in the original spectrum were collapsed to singlets, and subsequently, the algorithm identified the peak positions and heights from $Y_{2}$ in producing the WPT shift spectrum ($Y_{1}$). An automated sorting of molecules followed, where the peaks in $Y_{1}$ were matched with the library of the WPT shift spectra of pure molecules, which picked 64 molecules for what we call List I. In the next step, a design matrix, $X_{1}$, was created as per the description in Table 3, and the minimization of the quantity, $\sum {\left(Y_{1}-X_{1}\cdot \theta \right)}^{2}$, by a gradient descent was performed, where θ denotes the probability of the molecules in List I to be present in the mixture. The minimization was initiated by using a null vector of length 64 as θ. In this particular example, the minimization reduced the potential list of molecules to 62 (List II). In the next step, the top 15 molecules from List II were used to create another design matrix, $X_{2}$, and another gradient descent minimization of the quantity, $\sum {\left(Y_{2}-X_{2}\cdot \theta \right)}^{2}$, yielded an optimum θ. The final list of molecules corresponded to non-zero θ values, which in this case resulted in 8 molecules. The molecular composition matched the first 7 molecules in the prediction (true positives), while the last entry in the prediction was a false positive.

## 4. Conclusions

Composition analysis of small molecule mixtures is essential across a wide range of biological and organic research activities. While ${}^{1}$H NMR spectroscopy is a very powerful and effective technique in identifying small molecules, the NMR spectra of molecular mixtures are often poorly resolved due to spectral overlapping and the presence of multiplet structures. In this work, we presented an automated spectral analysis algorithm, which enhances spectral resolution by the application of the wavelet packet transform and predicts the associated molecular composition in a probabilistic manner. An augmented dataset of 1000 NMR spectra, corresponding to molecular mixtures containing 3 to 7 molecules, was used to test the efficiency of our method. We obtained a median true positive rate of 1.0 for all the mixtures with zero variation for the mixtures containing up to six molecules; the true positive rate for mixtures with seven molecules had a median and standard deviation of 1.0 and 0.08, respectively. A reasonably low false positive rate of 0.04 was achieved for the dataset. In addition, we demonstrated that the precision of the analysis could be further improved by visual inspection of the actual and predicted NMR spectrum of a molecular mixture, which can be automated as well. We believe that this method can enable high-throughput analysis of small molecule mixture compositions using ${}^{1}$H NMR as the primary or only spectroscopic tool.

## Figures and Tables

**Figure 1 molecules-28-00792-f001:**
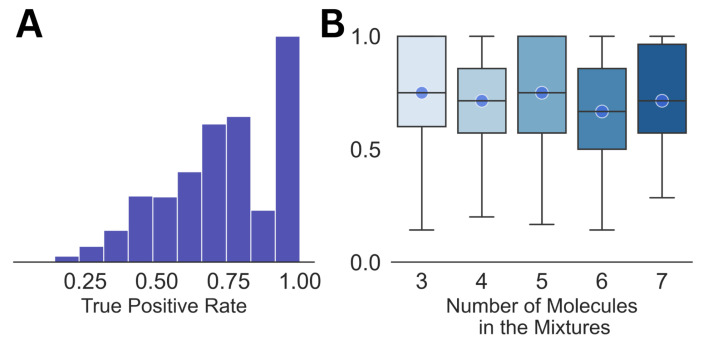
The distribution of the true positive rates for the entire dataset (**A**) and against the size of the mixture (**B**) is shown. The circles in (**B**) emphasize the median for each of the distributions.

**Figure 2 molecules-28-00792-f002:**
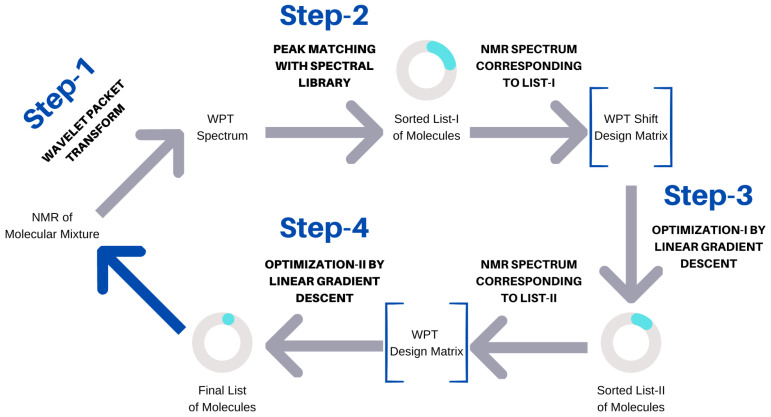
Schematic representation of the spectral analysis algorithm.

**Figure 3 molecules-28-00792-f003:**
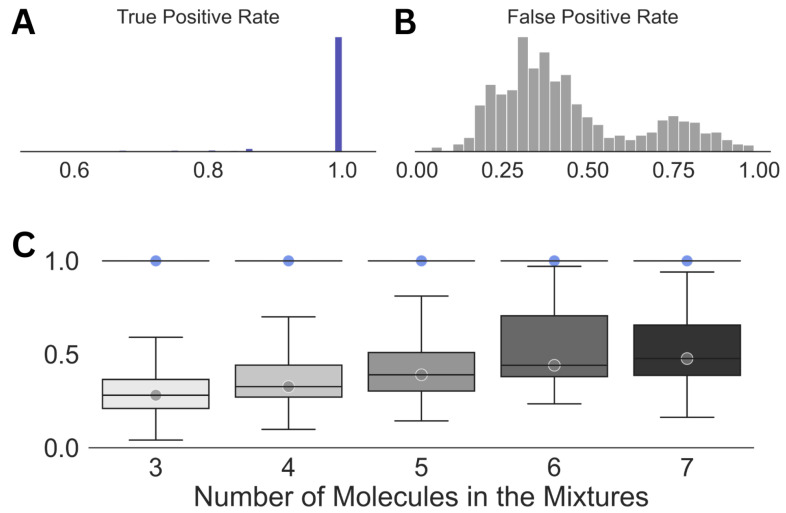
Summary of the results obtained in Step (3) of the analysis. Distributions of the true positive (**A**) and the false positive rates (**B**) for the entire dataset along with those against the size of the mixtures (**C**) are shown. The circles in (**C**) emphasize the median true positive rate (blue) and false positive rate (gray) for each of the distributions. For all the cases, a true positive rate of 1.0 was achieved (standard deviation = 0).

**Figure 4 molecules-28-00792-f004:**
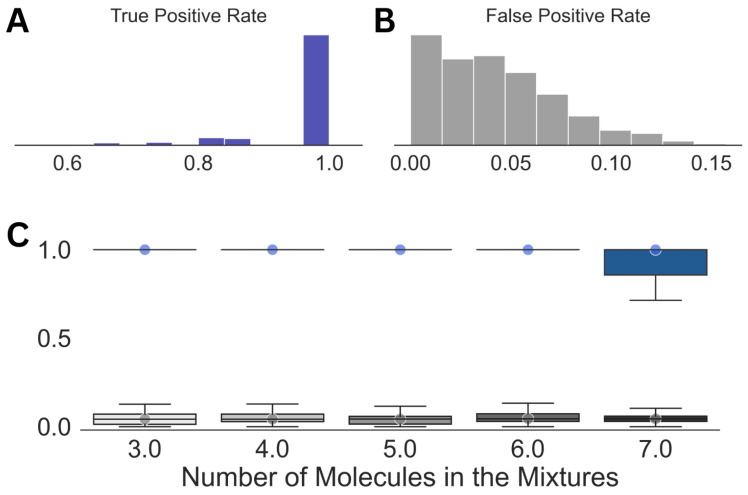
Summary of the results obtained after Step (4) in the analysis. Distributions of the true positive (**A**) and false positive rates (**B**) for the entire dataset along with those against the size of the mixtures (**C**) are shown. The circles in (**C**) emphasize the median true positive rate (blue) and false positive rate (gray) for each of the distributions.

**Figure 5 molecules-28-00792-f005:**
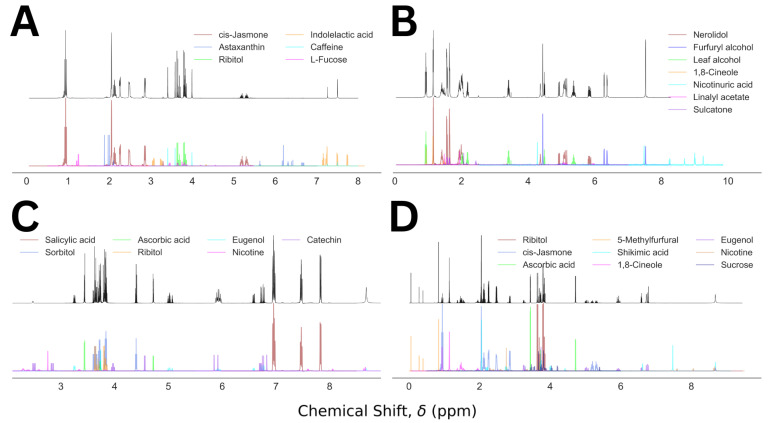
Mixed NMR spectra (black) and the predicted components (color coded) for Mixture Numbers 5 (**A**), 23 (**B**), 35 (**C**), and 20 (**D**), containing 3, 4, 5, and 6 molecules, respectively.

**Figure 6 molecules-28-00792-f006:**
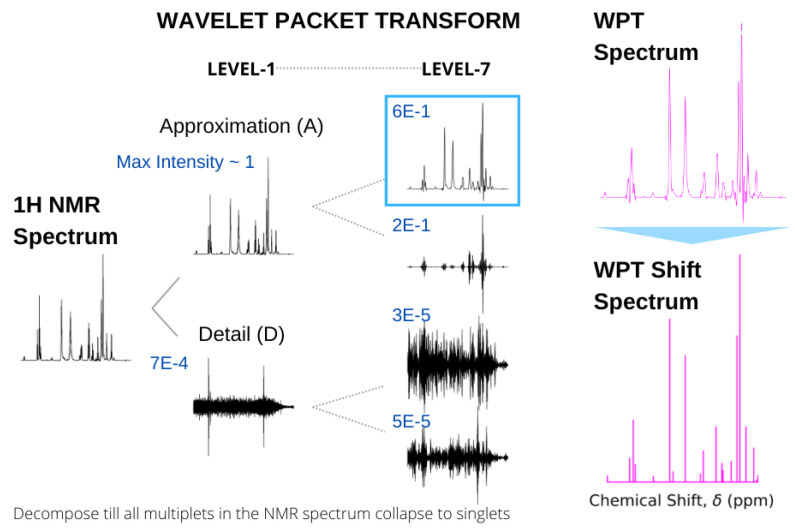
Conversion of the 500 MHz ${}^{1}$H NMR spectrum of glutathione (left) to WPT and WPT shift spectra (right). In calculating the WPT shift from the WPT spectrum, only the peaks above a threshold were taken into consideration. The wavelet decomposition at Level 1 and Level 7 by the Daubechies-9 wavelet (Db9) is shown, and the maximum amplitudes of each of the components are given in blue. A decomposition at Level 7 was selected, where all the multiplets in the original NMR spectrum were reduced to singlets.

**Figure 7 molecules-28-00792-f007:**
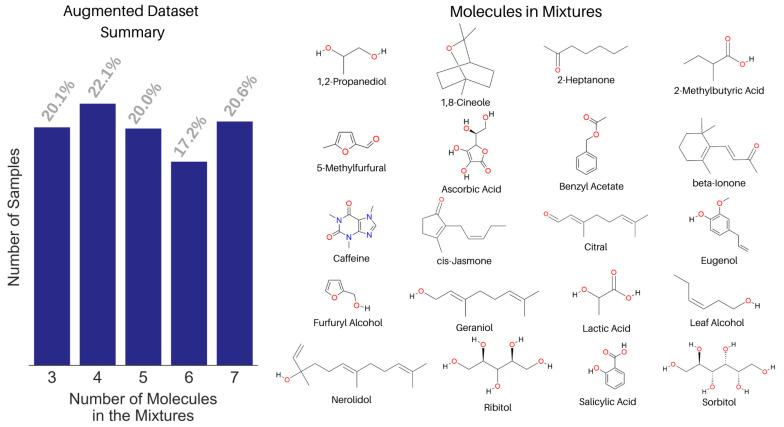
Summary of the augmented NMR spectral dataset with the fraction of samples against the number of constituent molecules in the mixtures (left) and the structure of the 20 molecules used in creating the augmented dataset (right).

**Figure 8 molecules-28-00792-f008:**
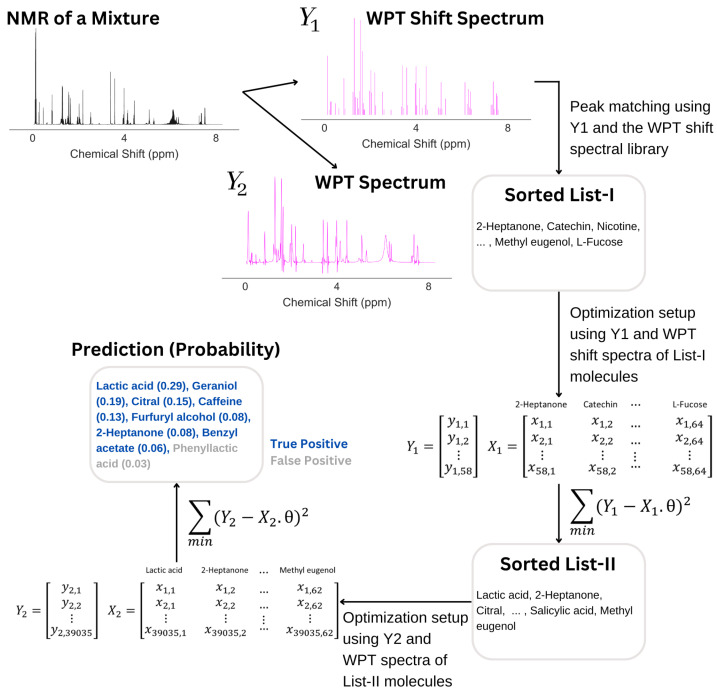
An illustration of how an NMR spectrum is analyzed in predicting the corresponding mixture composition. After calculating the WPT shift ($Y_{1}$) and WPT ($Y_{2}$) spectra from the NMR spectrum, an automated sorting selected 64 molecules (List I) from the library of 74 molecules by matching the WPT shift spectral peaks of $Y_{1}$ and that of the pure molecules from the library. An optimization of List I followed, yielding List II with 62 molecules. Another optimization of the top 15 entries from List II produced the final prediction, containing 8 molecules, with 7 of those corresponding to the true molecular composition of the mixture.

**Table 1 molecules-28-00792-t001:** Representative set of molecular mixtures and the corresponding prediction summary.

Mixture No.	Number of Molecules	Molecules (Proportions %)	True Positive Rate	False Positive Rate
5	3	Caffeine (39), ribitol (33), cis-jasmone (28)	1.0	0.04
23	4	Nerolidol (35), 1,8-cineole (22), leaf alcohol (22), furfuryl alcohol (21)	1.0	0.04
35	5	Sorbitol (28), eugenol (26), ribitol (18), ascorbic acid (15), salicylic acid (13)	1.0	0.03
20	6	Ribitol (20), eugenol (19), cis-jasmone (18), 5-methylfurfural (17), ascorbic acid (15), 1,8-cineole (12)	1.0	0.04

**Table 2 molecules-28-00792-t002:** Summary of the automated molecular mixture analyzer’s performance for the augmented NMR dataset.

Parameters	True Positive Rate	False Positive Rate
Mean	0.97	0.05
Median	1.0	0.04
Standard Deviation	0.09	0.03

**Table 3 molecules-28-00792-t003:** Calculation of a target *Y* and the corresponding design matrix *X*.

Chemical Shift	Target, *Y*	Design Matrix, *X*			
		${\mathrm{Molecule}}_{1}$	${\mathrm{Molecule}}_{2}$	*…*	${\mathrm{Molecule}}_{n}$
$\delta_{1}$	$y_{1}$	$x_{11}$	$x_{12}$	*…*	$x_{1n}$
$\delta_{2}$	$y_{2}$	$x_{21}$	$x_{22}$	*…*	$x_{2n}$
⋮	⋮	⋮	⋮	*…*	⋮
$\delta_{m}$	$y_{m}$	$x_{m1}$	$x_{m2}$	*…*	$x_{\mathrm{mn}}$

## Data Availability

The data used in this paper can be accessed via Signal Science Lab’s repository (7 December 2022) https://github.com/Signal-Science-Lab/Unsupervised_Molecular_Mixture_Analysis.

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
