# Peer review of "Unsupervised Analysis of Small Molecule Mixtures by Wavelet-Based Super-Resolved NMR"

_molecules, 2023, doi:10.3390/molecules28020792_

Round 1
Reviewer 1 Report
The authors aimed to resolve NMR resonances of mixture of small molecules using the wavelet packet transform (WPT) technique. In the previous study, they “decouple” multiplets in the NMR resonances. In this manuscript, the authors developed a spectral analysis to predict the composition of the corresponding molecular mixtures from their 1H NMR spectra in an automated fashion, by calculating WPT spectrum, conducting peak matching with a WPT shift NMR library, and predicting molecular composition of a mixture. The authors also tested robustness of the method using molecular mixtures, each containing 3 to 7 molecules. The study is scientifically excellent. However, I feel that the manuscript describes the method as a black box, which slightly lower the interest in the manuscript. Note that this is not for a NMR journal. Additional explanations to wider readership will be appreciated.
(1) Although the authors previously published WPT, the authors have to write a section the fundamental basis of the WPT based decoupling method, especially what information is used in the analysis. In reference 38, it is obvious that simplification of the multiplets is done by WPT, but it was not clear whether the intensity information is used to group the resonances for each compound.
(2) The authors have to explain more about the method. For example, what is “Targets (Y)” or “Target variable, Y1”? The entity of Y is unclear by the only description “the targets (Y) were taken to be WPT shift spectra for step-(3)”. Perhaps, examples of actual vectors and matrices will be helpful to understand the method.
(3) The authors have to compare whether the automated method using WPT and the commercial databased assignment method or comments on how the comparison will be.
Reviewer 2 Report
In this manuscript titled "Unsupervised Analysis of Small Molecule Mixtures by Wavelet-Based Super-Resolved NMR " - the methods of 1H NMR spectroscopy for spectral separation of small molecule mixtures is used. The contents of the reviewed manuscript are very well described by the title. The authors show that this method successfully predicted the spectral parts for small molecules with a median true positive rate.
In my point, the main message of this manuscript is very nice sound and could be recommended for publication in Molecules after a corresponding revision in which the following issues are addressed:
The innovation of this study was not enough. Please add more description and comparison for similar methods (DOSY NMR, RRCOSY, CORDY) used to solve this problem.
For example:
You can use just the DOSY coefficient for separate or different compounds in NMR spectra. It was demonstrated that CORDY could successfully separate the components with up to 2 orders of magnitude in the concentration dimension for the samples Yuan, B., Zhou, Z., Jiang, B., Kamal, G.M., et.al. NMR for Mixture Analysis: Concentration-Ordered Spectroscopy (2021) Analytical Chemistry, 93 (28), pp. 9697-9703. DOI: 10.1021/acs.analchem.1c00831
See another research about DOSY: Trefi, S., et.al., Martino, R. The usefulness of 2D DOSY and 3D DOSY-COSY 1H NMR formixture analysis: Application to genuine and fake formulations of sildenafil (Viagra) (2009) Magnetic Resonance in Chemistry, 47 (SUPPL. 1), pp. S163-S173. DOI: 10.1002/mrc.2490
Rogerson, A.K., et.al. Simultaneous enhancement of chemical shift dispersion and diffusion resolution in mixture analysis by diffusion-ordered NMR spectroscopy (2011) Chemical Communications, 47 (25), pp. 7063-7064. DOI: 10.1039/c1cc12456k
Another way you can apply relaxation time for separate signals NMR of different compounds.
Dal Poggetto, G.,et.al. Relaxation-encoded NMR experiments for mixture analysis: REST and beer (2017) Chemical Communications, 53 (54), pp. 7461-7464. DOI: 10.1039/c7cc03150e
Novoa-Carballal, R.,et.al. NMR methods for unraveling the spectra of complex mixtures (2011) Natural Product Reports, 28 (1), pp. 78-98. DOI: 10.1039/c005320c
or you can apply RRCOSY to separate signals for the same small molecules in the polymer matrix and free value. Sobornova, V.V., et.al. Molecular Dynamics and Nuclear Magnetic Resonance Studies of Supercritical CO2 Sorption in Poly(Methyl Methacrylate) (2022) Polymers, 14 (23), DOI: 10.3390/polym14235332
The authors have to shed light on the similarities and differences among their work and the literature of the problems for analysis of mixture analysis by NMR.
Round 2
Reviewer 2 Report
The necessary changes are correctly taken into account in new version, I propose to accept publications in present form.